# Faecal Glucocorticoid Metabolites and Hair Cortisone/Cortisol Measurements in Domestic Pigs Exposed to Road Transportation and Dexamethasone Treatment

**DOI:** 10.3390/ani14182700

**Published:** 2024-09-18

**Authors:** Camila J. Asencio, Rupert Palme, Héctor R. Ferrari, Mariano L. Lattanzi, Gabina V. Eguizábal, Juan M. Busso

**Affiliations:** 1Instituto de Investigaciones Biológicas y Tecnológicas (IIBYT), Consejo Nacional de Investigaciones Científicas y Técnicas (CONICET)-Facultad de Ciencias Exactas, Físicas y Naturales (FCEFyN), Universidad Nacional de Córdoba (UNC), Av. Vélez Sarsfield 1611, Córdoba X5016GCA, Argentina; camilaasencio@hotmail.com.ar (C.J.A.); gabina87@hotmail.com (G.V.E.); 2Instituto de Ciencia y Tecnología de los Alimentos, FCEFyN-UNC Av. Vélez Sarsfield 1611, Córdoba X5016GCA, Argentina; 3Department of Biological Sciences and Pathobiology, University of Veterinary Medicine, Veterinärplatz 1, 1210 Vienna, Austria; 4Cátedra de Bienestar Animal, Facultad de Ciencias Veterinarias, Universidad de Buenos Aires, Av. Chorroarín 280, Buenos Aires C1427CWO, Argentina; hferrari@fvet.uba.ar; 5Instituto Nacional de Tecnología Agropecuaria (INTA), EEA Marcos Juárez, Marcos Juárez X2580, Argentina; lattanzi.mariano@inta.gob.ar

**Keywords:** faeces, hair, non-invasive, glucocorticoids, pigs, road transport, physiology, reactive homeostasis

## Abstract

**Simple Summary:**

Transportation is one of the most stressful situations in pig production. The maximum duration of a journey should be determined considering the pigs’ coping capacity. The effect of a typical journey (200 km in Argentina) on pig physiological stress response was assessed using animal-friendly techniques. Fattening pigs (n = 18) were transported for three hours; 10 received a dexamethasone injection 7 h after the end of the journey. A control group (n = 18) was not transported, and 10 individuals received dexamethasone. The levels of faecal glucocorticoid metabolites and hair cortisol and cortisone were measured. The transported pigs had higher levels of faecal glucocorticoid metabolites after transportation than the control group. The level of faecal glucocorticoid metabolites returned to normal values faster in transported pigs treated with dexamethasone than in the control group. The amount of glucocorticoids in their hair was not related to the stressful stimuli; in contrast, the cortisone/cortisol ratio yielded better results. Since glucocorticoid measurements in hair are still a questionable stress biomarker in pigs, further studies are needed to develop animal-based transport protocols. The results showed that pigs were able to cope with a 3 h road trip. In the near future, a non-invasive glucocorticoid analysis could be used to characterize pigs’ coping capacity on a farm.

**Abstract:**

Pig homeostasis is challenged by stressful production practices, like road transportation. Glucocorticoids (GCs) are mediators of reactive homeostasis, and their concentrations are frequently used as a stress indicator. The adrenocortical activity of fattening female and castrated male pigs was monitored over a 5-day longitudinal study. A bi-factorial experimental design was applied on day 2; 18 pigs in pen 1 were transported for 3 h (T; 1.2 m^2^/pig), and 18 pigs were kept in pen 2 (NT). Ten pigs from each pen were treated with dexamethasone (T-D or NT-D), and eight with saline solution (T-SS or NT-SS). Adrenocortical activity was assessed by measuring the levels of faecal glucocorticoid metabolites (FGMs) and hair cortisol and cortisone. In T-SS pigs, the level of FGMs was higher after transportation than in NT-SS pigs. The level of FGMs of T-D pigs initially increased but then reached similar levels to those of NT-SS sooner than T-SS. In contrast, hair cortisol and cortisone did not respond to the treatments. Nevertheless, the hair cortisone/cortisol ratio increased due to transport and decreased after dexamethasone administration. Daily faecal sampling proved still more reliable than 60-day hair sampling for assessing adrenocortical activity. Transported pigs recovered their adrenocortical baseline levels within 24 h. Dexamethasone attenuated the response to transport.

## 1. Introduction

Domestic pigs may be transported several times during their life for production and market reasons [1]. The transportation process of fattening pigs can cause economic losses due to mortality, skin damage, and the general deterioration of meat quality [2,3]. The World Organization for Animal Health (WOAH) [4] recommends some animal welfare principles, such as a maximum duration of a journey, which should be determined by the ability of the animals to cope with the stress of transport (article 7.3.5: planning the journey). In countries such as Argentina, where pigs travel long distances across the country (up to 2000 km; 63.51% travelled up to 200 km in 2018) [5] from the farm to the slaughterhouse, the distance pigs can cope with should be determined based on an animal-based measurement. Endocrine biomarkers of stress are being increasingly employed in pig welfare studies [6,7,8,9,10]. Animal-friendly techniques, such as non- or minimally invasive ones, may be useful to explore this topic. In this sense, studies on non-invasive glucocorticoid (GC) measurements in pigs have increased (e.g., [11,12,13,14,15,16,17,18,19]). Still, studies under controlled conditions, whether in research facilities or farms, are necessary to support the use of these animal-friendly techniques in pigs with the aim to define transport protocols.

In vertebrates, the hypothalamus–pituitary–adrenal (HPA) axis plays a key role in helping to cope with stressors through the secretion of glucocorticoids (GCs) [20]. Hence, these hormones are important mediators of the physiological stress response, and their concentrations are frequently used as an important stress indicator. The quantification of GC metabolites in faeces (FGMs) has been used as a non-invasive method to assess stress in several species [21]. Faeces can be collected easily without stressing the animal [22], thereby enabling repeated GC metabolite measurements in individuals [23]. In faecal samples, circulating GC concentrations are integrated over a certain period and represent the cumulative secretion of GCs compared to point estimates obtained from blood samples. Therefore, faecal samples are less affected by short episodic fluctuations or the pulsatile nature of GC secretion [23,24]. Thus, this non-invasive approach is more useful to assess the welfare status and the change induced by management stressors than GC measurement in blood [25].

In recent years, the use of hair cortisol as a biomarker of physiological stress has been explored in different mammalian species [26,27,28,29,30], including pigs [11,13,31,32]. Storing hair samples is easier and would allow for a retrospective analysis of endogenous cortisol exposure [33,34]. In addition, cortisol is inactivated in the hair follicle by the 11β-hydroxysteroid dehydrogenase (11β-HSD) type 2 enzyme, converting it to cortisone [35,36]. Although hair cortisone has received little attention so far, previous studies have indicated that it may be a useful additional biomarker for stress research in this biological matrix [36,37,38,39,40]. Moreover, recent studies on pigs have focused on the combination of these two main glucocorticoids in saliva. For example, a greater increase in the cortisone/cortisol ratio than in cortisol was observed during periods of high stress; this result was attributed to an increase in the activity of 11β-HSD type 2 [19,41]. The sample to measure GC in hair as a biomarker of stress should contain enough actively grown hairs, which can be achieved by the “shave–reshave” method [14]. Heimbürge et al. [42] found that hair growth on pigs’ necks can reach up to 5 mm/month. Therefore, we chose an interval of several weeks to ensure that enough hair exposed to systemic cortisol levels modulated by the treatments could be harvested. Nevertheless, GC quantification in hair samples to monitor endocrine stress still does not provide accurate evidence of the expected cause and effect changes [43].

Several studies have addressed the response of farm animals to transport [44,45,46,47,48]. However, to our knowledge, no study has focused on FGM and hair GC response of pigs to a situation that activates or inhibits the HPA axis (in our case, transport and dexamethasone, respectively), and the synergistic effects of endogenous and exogenous GCs on HPA dynamics.

The aim of the present study was to assess FGMs and cortisone/cortisol in hair as possible indicators of adrenocortical dynamics in pigs exposed to two types of stimuli: transport (HPA stimulation) and dexamethasone injection (HPA suppression), as biological and pharmacological factors, respectively. This study was developed considering the framework of the Reactive Scope Model proposed by Romero et al. [49] and applied by our research group [50], focusing on reactive homeostasis. Reactive homeostasis is the range of concentrations/levels of mediators needed to respond to unpredictable or threatening environmental changes, in our study, transport or dexamethasone. We predict that pigs exposed to an acute stimulus such as transport will be able to reach baseline FGM values within a short period and that synthetic GCs such as dexamethasone will induce negative feedback effects on HPA activity by reducing natural GCs. We also predict that transported pigs treated with dexamethasone will exhibit different adrenocortical dynamics and that baseline FGM values will be reached earlier than in control pigs due to the synergistic effect of endogenous GCs (transport) and synthetic GCs (dexamethasone) on the negative feedback mechanism of the HPA axis. Overall, we predict a positive correlation between GC in hair and FGM changes, and we expect a higher cortisone/cortisol ratio in pigs exposed to transport than the pigs in the control or dexamethasone groups. Vitousek et al. [51] suggest focusing on the variation in the ability to terminate the stress response of vertebrates through negative feedback as an important component of the animal’s stress coping capacity. Hence, we consider that pigs’ ability to rapidly and effectively terminate the short-term response to stress may be fundamental to cope with dynamic environments, such as commercial farms and the slaughterhouse.

## 2. Materials and Methods

### 2.1. Study Pigs and Housing Conditions

Thirty-six crossbred (Landrace x Yorkshire x Pietrain) fattening pigs (n = 18 females and n = 18 castrated males) were studied under intensive breeding conditions at the research unit of the National Institute of Agricultural Technology (INTA), Marcos Juarez, Córdoba, Argentina (32°42′60.0″ S; 62°06′02.8″ W). The Pietrain sire terminal line used was homozygous halothane-dominant, i.e., a halothane-free sire line. This study was carried out from November 2022 to January 2023 (spring-summer) under semi-controlled environmental conditions (natural changes in photoperiod and temperature). Balanced feed and water were provided ad libitum. The average body weight was 53.5 ± 1.1 kg in November and 97.9 ± 1.8 kg in January.

Pigs were randomly assigned based on sex and body weight to one of two contiguous study pens located in the same building (pens 1 and 2, 3.5 × 5.2 m in size, 0.8 m^2^/pig; sex ratio 1:1). Under a breeding-productive condition, the number of pigs per pen determined the sample size for the experiment (see below), which was suitable with the criteria n > p, where n = number of pigs and p = number of repeated measurements. The pens had a fully slatted concrete floor. The pigs were weighed before the beginning of the study to balance the groups for body weight.

### 2.2. Experimental Design

Each pig was marked at weaning with a numbered tag in the right ear. For an easier identification, the same tag number was marked with synthetic spray paint on the pigs’ back (close to the tail). A bi-factorial experimental design was applied on day 2 of a 5-day longitudinal study. The pigs in pen 1 were assigned to the transported group (treatment 1) and those in pen 2 to the not transported group (control 1). Treatment 2 was applied to some of the pigs in both pens, which were assigned either to the dexamethasone group (treatment 2: dexamethasone) or to the saline solution group (control 2). The treatments and controls are summarized in Table 1.

### 2.3. Treatment 1: Road Transportation

The pigs (n = 9 females and n = 9 males) were transported in a truck for 3 h. The journey was authorized by the local authority of SENASA (National Service for Agri-Food Health and Quality, authorization #: 1-344430). Road transportation was carried out in the morning (08:00–11:00 a.m.) of day 2. Before loading, the suitability of the pigs for transportation was confirmed in accordance with the provisions of SENASA, Resolution 1697/2019. All the transported pigs were able to move without assistance and none had open wounds or prolapses. They belonged to the same pen and did not have physical contact with pigs from the other pen. The pigs had free access to food and water prior to transport. Only one herding panel was used to facilitate the movement of the pigs from the pen to the truck through a chute (approximately 50 m long). The pigs were loaded onto and unloaded from the truck using a wooden ramp. Loading and unloading took approximately 6 min.

The truck had a single compartment for the pigs (1.2 m^2^/pig) and a metal grid floor; there was no roof or any type of aerial cover. During transport, the pigs did not have access to food or water. The duration of transport was chosen according to data published by SENASA [5]. Paved routes were used, and traffic regulations were followed (travelling speed range: 60–80 km/h). On the day of transport, the weather was as follows: clear sky, temperature 25.7 °C, humidity 41.7%, and 69.8 on the temperature–humidity index (THI) at the beginning of the transport (08:00 a.m.). At the end (11:00 a.m.), temperature was 34.0 °C, humidity 25.1%, and 78.3 THI (data were collected from https://new.omixom.com/next/station (accessed on 15 May 2023). The control group consisted of not transported pigs (n = 9 females and n = 9 males) that were kept in the pen with access to feed and water.

### 2.4. Treatment 2: Dexamethasone Injection

Twenty pigs (n = 5 females and n = 5 males of each pen—transported and not transported pigs) were injected with a single dose of dexamethasone (i.m. 0.36 mg/kg). The dose was calculated based on a previous study (see Appendix A). Dexamethasone was administered after transportation in the afternoon (06:00 p.m.) of day 2.

The control group consisted of 16 pigs, 8 pigs (n = 4 females and n = 4 males) from each pen (transported and not transported pigs), injected with a single dose of saline solution (SS; NaCl 0.9%; i.m. 5 mL/pig).

### 2.5. Sample Collection and Steroid Extraction

For the FGM measurements, fresh faeces were collected daily immediately after defecation by two operators per shift, from 6:00 to 12:00 and 13:00 to 19:00 h, across 5 days. The samples (total number of samples: 430) were identified by the tag number of the pig and stored at −20 °C until analysis. The time at which each sample was collected was recorded and included in the data analysis. The FGMs were extracted following a simple method: 5 mL of methanol/water (80%) was added to a portion (0.5 g) of each well-homogenized sample [52]. After shaking (2 min) and centrifugation (15 min; 3000× *g*), an aliquot (0.5 mL) of the supernatant was separated for further use. The extracts were evaporated at 60 °C and sent to the laboratory in Vienna.

To measure GCs in hair, at the end of the study (after 60 days of treatment), a white hair sample was collected from each pig by reshaving a previously (7 days before the start of the experiment) shaved area located in the pig’s back (between shoulders, approximately 10 × 10 cm) with a hair clipper, trying not to remove the root of the hair. The shaved hair length was approximately 10 mm, which included the hair grown (or exposed) during the experimental period. The hair sample was stored at −20 °C until analysis (total number of samples: 36). A portion of 0.2 g of each hair sample was washed with 7 mL of n-hexane (100%) and shaken for 1 min using a manual vortex. The n-hexane supernatants were discarded, and the hair samples were dried. For extraction, a portion (0.1 g) of washed hair was immersed in 5 mL of 100% methanol and incubated at 37 °C for 24 h with gentle rotation (Thermomixer, stage 5 of 10, Eppendorf, Hamburg, Germany). After centrifugation (15 min; 2500× *g*), an aliquot (2.5 mL) of the supernatant was evaporated at 60 °C and sent to the laboratory in Vienna.

### 2.6. Measurement of Faecal Glucocorticoid Metabolites

The faecal extracts were resuspended in 80% methanol and diluted in enzyme immunoassay (EIA) buffer (1 + 9). All samples were measured in duplicate with a 5α-pregnane-3ß,11ß,21-triol-20-one EIA (for details, see [53]). This assay is a group-specific EIA, which measures metabolites with a 5α-3ß,11ß-diol configuration. It was previously successfully validated for domestic female pigs [12]. The sensitivity of the EIA was 4 ng/g faeces. The intra-assay coefficient of variation was always below 10%. The values are expressed as ng/g faeces.

### 2.7. Measurement of Hair Glucocorticoids

The hair extracts were resuspended in 0.5 mL of the EIA buffer (1:10). All measurements were taken in duplicate using two EIAs: (1) cortisol EIA (for details, see [54]) and (2) cortisone EIA, as described Rettenbacher et al. [55]. The sensitivities of the EIAs were 0.2 and 0.4 ng/g hair, respectively. In all cases, the intra-assay coefficient of variation was below 10%. The values are expressed as ng/g hair, and the cortisone/cortisol ratio was calculated.

### 2.8. Statistical Analysis

A baseline FGM value was obtained for each individual from the faecal samples collected in the morning of day 1 before treatment. Next, the percentage of each value relative to the baseline was calculated. To avoid possible circadian effects, the values obtained from the samples taken in the morning and afternoon of each day (days 2, 3, and 4) were averaged, except for the treatment day.

The values of the FGMs (%) were log-10 transformed to meet normal distribution, and a linear mixed model (LMM) was applied. The levels of fixed factors were transport and no transport for treatment 1; dexamethasone and saline solution for treatment 2; and pre-treatment (d 1, am), day 1 (d 1, pm), day 2 (d 2, am + d 2, pm), day 3 (d 3, am + d 3, pm), and day 4 (d 4, am + d 4, pm) for time. The random factors were individuals and sex.

The hair data were also log-10 transformed to meet normal distribution, and an analysis of variance (ANOVA) was applied to determine the statistical differences between treatments (treatment 1: transport and no transport; treatment 2: dexamethasone and saline solution), considering the interaction between factors.

All analyses were performed using InfoStat [56]. The values are reported as mean ± SEM unless otherwise indicated, and the significance level was 5% for all tests. Normality was checked using the modified Shapiro–Wilk test and a Fisher’s a posteriori test was applied when the statistical analysis showed a *p* ≤ 0.05.

## 3. Results

### 3.1. Measurement of Faecal Glucocorticoid Metabolites

The statistical analyses detected an effect among treatments 1 and 2, and time on percentage of FGM change (F_3,107_ = 2.71; *p* = 0.0486). In the control group (NT-SS), the FGM values decreased during the experimental period. In the T-SS group, the values were similar to those of the control group on day 3. In the pigs of the NT-D group, the FGM values decreased after day d 1 (pm), when dexamethasone was applied, but no differences from the control group were detected. The pigs of the T-D group showed a similar profile to that of the T-SS and NT-D groups on day 1; then, the T-D group reached similar values to those of the control group on day 2 until the end of the study (Figure 1).

### 3.2. Measurement of Glucocorticoids in Hair and Cortisone/Cortisol Ratio

The statistical analysis of cortisol concentrations in hair showed significant differences for the transported pigs (F_1,2.42_ = 66.67, *p* < 0.0001). The a posteriori test showed that the not transported group (NT + SS and NT + D) had the highest value. For cortisone, an effect of dexamethasone was found (F_1,0.93_ = 9.01; *p* = 0.0052), with the lowest values being detected in the injected pigs (NT + D and T + D). A more detailed description of the results is depicted in Figure 2, showing the values for each treatment.

The ANOVA showed that the hair cortisone/cortisol ratio in the pigs changed significantly in response to transport (F_1,2.43_ = 24.28; *p* < 0.0001; Figure 3) and dexamethasone (F_1,0.74_ = 7.37; *p* = 0.0106; Figure 4).

## 4. Discussion

In this study, three different EIAs were used to monitor the adrenocortical response to road transportation and dexamethasone challenge in the faecal and hair samples of fattening pigs reared under intensive conditions in a research unit (similar to farm facilities) in Córdoba, Argentina. The EIAs have been tested in several mammalian species to obtain information about a physiological stress indicator such as adrenocortical activity [22,25,57]. Our findings indicate that the pigs were able to cope with the biological challenge (a 3 h journey). Moreover, although dexamethasone injection alone did not affect FGM concentration, this treatment combined with transport treatment affected adrenocortical activity. Thus, dexamethasone administration after transport facilitated a quicker recovery of adrenocortical dynamics in the studied pigs. Neither the cortisol nor cortisone EIAs were suited to demonstrate the expected changes in adrenocortical dynamics when applied separately to 60-day regrown hair samples. In contrast, estimates of the cortisone/cortisol ratio confirmed our predictions, showing higher values after road transportation and lower values after dexamethasone administration.

The analysis of FGMs revealed that adrenocortical activity increased after 3 h of road transportation, with the highest activity being recorded by 12–24 h after the end of the journey. This result confirms that FGMs can be used to assess physiological stress reactions not only in female domestic pigs but also in castrated male domestic pigs (e.g., [12,25]). Considering that both sex and castration may influence cortisol release in mammals, and thus measurement in faeces [21] and hair [17], future assessments should take these factors into account to address individual variability in the physiological stress response. This finding also indicates that the adrenocortical axis required 24 to 48 h to return to baseline levels after the end of the 3 h truck journey. Pigs non-invasively monitored in the present study were able to cope with the duration of this journey, as indicated by the physiological biomarker used. On the contrary, Werner et al. [58] found that not only very long (8 h) but also short (1 h) journeys affected other welfare indicators, particularly in summer. In Europe, the regulations on animal welfare during pig transport to slaughter focused only on long-term journeys. However, the results from short journeys also indicated an effect on the welfare of the pigs [58]. In the context of slaughterhouses, the present results mean that adrenocortical activity may increase during the last few hours before sacrifice. Therefore, the influence of a second multidimensional stressor, like a new environment (a slaughterhouse), on transported pigs would increase adrenocortical activity, threatening reactive homeostasis. The addition of several important stressors may elevate GC levels above the reactive homeostasis threshold, i.e., a homeostatic overload may generate failure in pig health, negatively affecting meat quality (framework hypothesis postulated by [49]).

Dexamethasone (treatment 2) did not affect adrenocortical activity in fattening pigs, according to FGM measurements. This result was unexpected, since this artificial GC usually reduces blood cortisol concentration, and we expected a lower level of adrenocortical activity than in the pigs of the control group. In a former study conducted in the same research unit (see Appendix A), dexamethasone administration reduced adrenocortical activity in fattening pigs. The inconsistencies between the present results and those reported in the appendix may be due to differences in the number of dexamethasone administrations (three injections in the treatment reported in the Appendix A vs. one application in the present study). When steroids are measured in excreta, steroid metabolites represent a pooled fraction of excreted hormones, providing an integrated measure of steroid level over a longer period than steroid concentration in the blood does [59]. Possibly, a single administration may have generated a short-term effect in the bloodstream that might not be detectable in faeces, since pigs have low defecation rates [59].

The combination of treatments clearly showed that adrenocortical activity initially increased in transported pigs during the afternoon and decreased on the following day, reaching FGM values of the control group. In this sense, the pigs were exposed to transport and dexamethasone, two antagonistic stimuli that produce the activation and inhibition of the HPA axis, respectively. Thus, we found a synergistic relationship between dexamethasone and endogen GCs released during transport, with the consequent negative feedback effect on HPA. The transported pigs that did not receive dexamethasone showed higher levels of adrenocortical activity than those that received dexamethasone. This finding may support the hypothesis proposed by Vitousek et al. [51], who postulated that the ability of vertebrates to rapidly and effectively terminate the short-term response to stress might be fundamental to surviving in dynamic environments.

The analysis of hair GCs and the cortisone/cortisol ratio in pigs revealed results that deserve attention. Although the diffusion of GCs from blood or body secretions (sweat and sebum) into hair has been proposed [34], new studies have suggested that hair GCs are mainly synthesized in the hair follicle [60]. Hair follicles were reported to contain a functional equivalent of the HPA axis and can synthesize cortisol after stimulation by a corticotrophin-releasing hormone [61,62]. Studies performed in animals show that a wide array of stressors and pathological conditions altered cortisol concentrations in hair. However, further research is necessary to understand the underlying mechanisms of cortisol incorporation into hair and to explore the hair growth characteristics in the species of interest. To overcome confounding influences, the use of standardized sampling protocols is strongly recommended [14].

In general, a hair sample may integrate GC levels over a longer period than a faecal sample. In accordance with a previous study measuring cortisol and cortisone in the blood samples of pigs [62], we found that in the hair samples of the control group, the cortisol levels were higher than the cortisone levels (Figure 2). Based on the applied treatments, we expected higher GC concentrations in the transported pigs than in the not transported pigs and/or the pigs treated with dexamethasone. The results of the present study showed that the cortisol EIA was not useful for detecting the expected changes. This finding is line with those reported by Wiechers et al. [63], who compared farrowing systems with different situations of chronic stress using pig hair samples and did not find significant differences in hair GCs. We also used a cortisone EIA considering that cortisol is inactivated in hair by the action of the 11β-hydroxysteroid dehydrogenase (11β-HSD) type 2 enzyme, converting it to cortisone [28,36], as indicated in studies involving humans and sheep [37,40,64]. Moreover, cortisone is less polar than cortisol; therefore, the increased incorporation of cortisone into hair from the bloodstream would be expected [37]. However, in the present study, we were not able to show the expected differences using the cortisone EIA.

Cortisol and cortisone are two main glucocorticoids involved in the stress response, and they are also reported in combination [65]. In sheep, the topic was explored and more studies elucidating the incorporation of cortisol and cortisone into hair are needed to support this approach [40,64]. We found higher cortisone/cortisol ratios in transported pigs than in not transported ones, and dexamethasone-treated pigs exhibited lower ratios than the ones injected with the saline solution. Escribano et al. [66] demonstrated that pigs subjected to a high atmospheric temperature (a typical environmental stressor) exhibited changes in adrenocortical activity compared to pigs kept in a cooling room on the farm and that the cortisone/cortisol ratio was more efficient than cortisol alone. In that study, as well as in the present study, pigs were exposed to a non-cooling environment and were under thermal stress on the farm, as indicated by the reported values of THI > 74. Escribano et al. [66] pointed out that it is preferable to use cortisone and the cortisone/cortisol ratio (an estimate of 11β-HSD type 2 activity) as indicators of chronic heat stress in pigs. Furthermore, the use of the ratio may also help to discriminate systemic GC from local GCs. After injecting radiolabeled cortisol in guinea pigs, Keckeis et al. [60] found radiolabeled cortisone but not cortisol in the hair. This result underlines that systemic cortisol is inactivated in the hair follicle into cortisone. Therefore, cortisone may be better suited for measuring overall GC levels. In contrast, unlabeled cortisol was still found in those hairs (measured by EIA), indicating local production [60]. Considering the present results, the increased ratio in transported pigs could be explained by a high level of blood cortisol that is converted and incorporated into the hair as cortisone. On the contrary, the decrease in the ratio after dexamethasone treatment is explained by the low blood cortisol level due to the HPA axis suppression. Additionally, since dexamethasone may be “picked up” by the cortisol EIAs, any increase in cortisol due to cross-reactions would contribute to an even greater reduction in the ratio. Although the cortisone/cortisol ratio was a useful indicator of stress changes after road transportation and dexamethasone challenge in the present study, further studies are needed to confirm its usefulness as a robust indicator to develop protocols in the context of the World Organization for Animal Health.

The unexpectedly high values in hair cortisol detected in the control group compared to the values detected in the transported pigs, as revealed by the cortisol EIA, may be attributed to different confounding factors. We speculated that the hair sampling period and/or the climatic conditions, rather than a lack of effect of biological and/or pharmacological challenges, would have a confounding effect on our findings. On the one hand, the 60-day period used in the present study was longer than the sampling period used by Heimbürge et al. [29]. These authors found no changes in hair cortisol concentrations in pigs at around week 4 after repeated ACTH administration. The authors pointed out that a low systemic cortisol response in pigs might explain the lack of changes in the measurements of cortisol in hair. They also indicated other sources of variation, such as seasonally reduced hair growth and hair contamination, which may interfere with the validity of hair cortisol. We ruled out contamination problems because we collected hair samples from the neck, and the pigs were not dirty. We also ruled out low availability of systemic cortisol, since we collected hairs probably in the anagen phase. We used the shave–reshave technique [67], ensuring that the collected samples included hair only in active growth phases before the sampling period. This approach may provide more accurate evidence of systemic cortisol concentrations over the preceding weeks than natural hair, as elucidated by Heimbürge et al. [14]. On the other hand, the pigs were exposed to direct sunlight during transport because the truck did not have a roof or any type of aerial cover. When the pigs arrived at the pen, we detected red skin (probably inflammation due to sun exposure). Previous reports indicated that inflammation fosters an anagen-to-telogen transition and that ultraviolet exposure can lead to early teloptosis (exogen phase) [68,69]. Recently, shaved hair and/or hair follicles have been shown to possibly be negatively affected by the sun, damaging the health of hair and reducing the incorporation of cortisol from the bloodstream during the following days. Moreover, Otten et al. [70] demonstrated that artificial light irradiation degraded hair cortisol in vitro. This may explain the lower hair cortisol concentration in the transported pigs exposed to direct sunlight compared to the not transported pigs. Misinterpretations associated with seasonality or seasonal climatic variation in different regions have also been reported as a source of variation in the application of this methodology (e.g., [71,72,73]). Thus, further studies are necessary to support the use of this tool in pigs, focusing on different sampling periods of reshaved hair collection and/or on the stress response of pigs exposed to controlled environmental conditions to avoid misleading factors, such as climatic conditions.

Finally, the present study supports the use of FGM concentrations as a reliable indicator of stress in pigs. Non-invasive hormone monitoring, a very useful research laboratory tool still not used as a standard method in veterinary clinics, is labor-intensive and requires many samples to evaluate the effects of stressors. Studies evaluating cortisol metabolites in regularly collected faecal samples during and after transportation have proven valuable in other species [74,75,76]. On the other hand, in the context of animal production, hair has several advantages over other biological matrixes used in endocrinology laboratories, since it can be collected with minimally invasive methods, and can be easily transported and stored, and, importantly, only one sample may be necessary for diagnosis. However, the cause-and-effect relationship between stress and elevated GC levels in hair, sampled weeks later, is hard to prove. Kalliokoski et al. [43] pointed out that in controlled facilities, like the one used in the present study, hair glucocorticoid measurement seems to be valid for adrenocortical monitoring, and they are positively correlated with measurements in faeces. This prediction was not confirmed in the present study. Nevertheless, the hair cortisone/cortisol ratio was positively correlated with the faecal glucocorticoid measurements. Further studies are needed to validate this ratio as a better indicator of stress in pigs than individual measurements of hair cortisol and cortisone.

## 5. Conclusions

The present results indicate that non-invasive monitoring of adrenocortical activity in fattening pigs could be used to assess the effect of a stressor such as transportation, a multifactor stimulus that differs from a single-factor and acute stimulus like dexamethasone injection. Pigs subjected to 3 h of transportation exhibited a higher level of adrenocortical activity during 24 h compared to not transported pigs. The administration of dexamethasone after transportation clearly accelerated the recovery of adrenocortical activity to baseline levels compared to the transported pigs that did not receive dexamethasone.

Since hair cortisol and cortisone did not respond to the treatments, the comparison of FGMs and cortisol and cortisone in hair needs further studies to explore the suitable timing of sample collection. Here, a 60-day window was apparently unsuitable for measuring GCs with the applied EIAs. However, the calculated cortisone/cortisol ratio yielded better results. Therefore, more studies are also necessary to elucidate the relationship of these two main glucocorticoids present in pig stress response, and the use of these measurements in the hair of pigs to contribute to the design of transport protocols.

## Figures and Tables

**Figure 1 animals-14-02700-f001:**
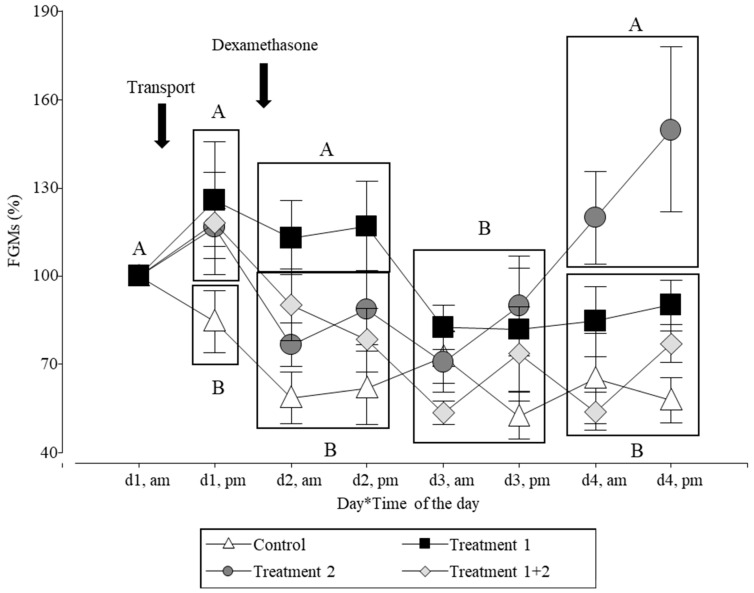
Changes in the FGM levels of the pigs relative to the concentration on d1, am before the treatment. The pigs (n = 36) were housed under intensive farming conditions. The arrows indicate the day of the treatments. Control (NT + SS; n = 8 pigs): not transported pigs injected with saline solution. Treatment 1 (T + SS; n = 8 pigs): transported pigs injected with saline solution. Treatment 2 (NT + D; n = 10 pigs): not transported pigs injected with dexamethasone. Treatment 1 + 2 (T + D; n = 10 pigs): transported pigs injected with dexamethasone. Different letters indicate significant differences between days (the values of am–pm of each day were averaged = rectangle, except for day 1) (*p* ≤ 0.05).

**Figure 2 animals-14-02700-f002:**
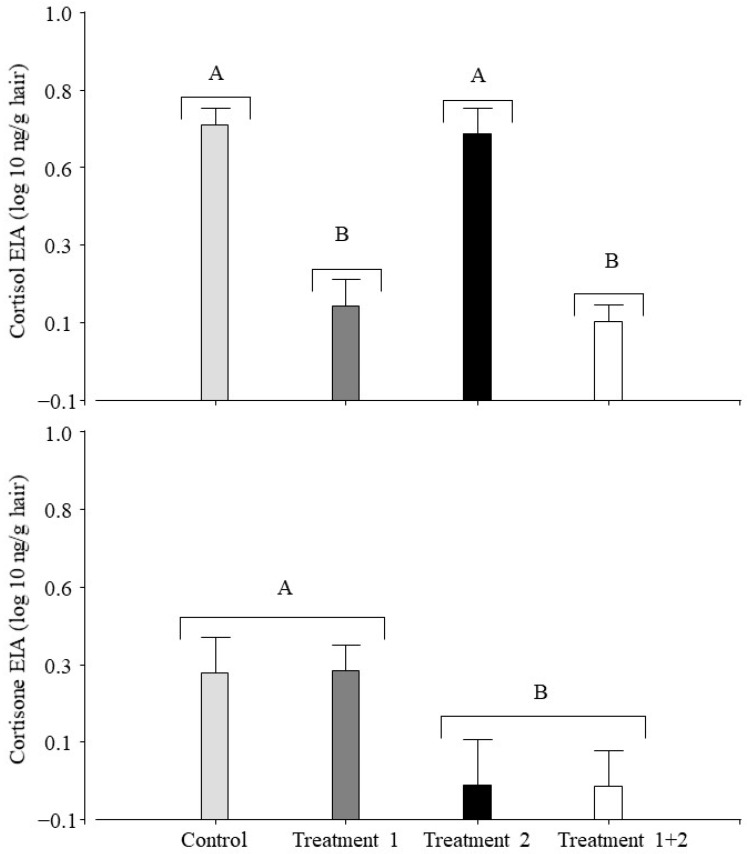
Effect of treatments (road transportation and dexamethasone) on hair glucocorticoids (measured using two EIAs) of pigs (n = 36) housed under intensive farming conditions. Control (NT + SS): not transported pigs injected with saline solution. Treatment 1 (T + SS): transported pigs injected with saline solution. Treatment 2 (NT + D): not transported pigs injected with dexamethasone. Treatment 1 + 2 (T + D): transported pigs injected with dexamethasone. Each panel represents the results using one of the two EIAs. Sixty days after the end of the study, a hair sample was collected by reshaving the previously shaved area located at the pig’s back. Different letters indicate significant differences within each panel (*p* ≤ 0.05).

**Figure 3 animals-14-02700-f003:**
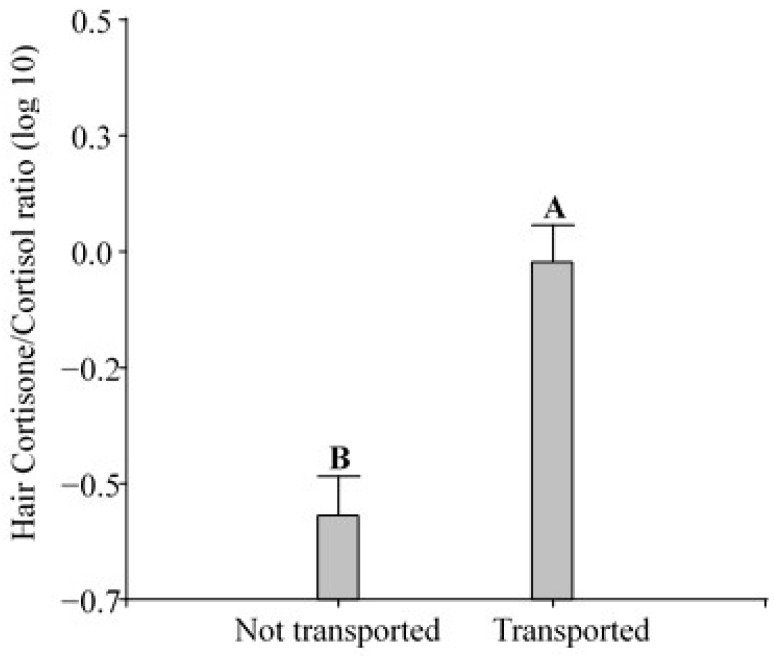
Effect of road transportation on hair cortisone/cortisol ratio in pigs (n = 36) housed under intensive farming conditions. Both the not transported and transported groups included pigs that were injected either with dexamethasone or a saline solution. Different letters indicate significant differences (*p* ≤ 0.05).

**Figure 4 animals-14-02700-f004:**
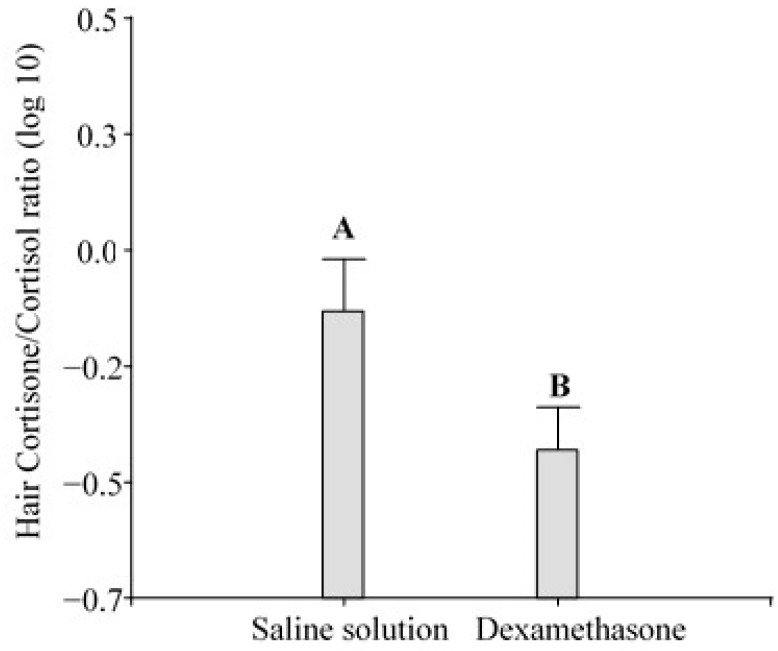
Effect of dexamethasone on hair cortisone/cortisol ratio in pigs (n = 36). Both the saline solution and dexamethasone groups included pigs that were either transported or not transported. Different letters indicate significant differences (*p* ≤ 0.05).

**Table 1 animals-14-02700-t001:** Experimental groups of pigs randomly assigned to different treatments.

		Treatment 1		
		Transported (T)	Not Transported (NT)	
Treatment 2	Dexamethasone (D)	5♀ & 5♂ = treatments 1 + 2 group	5♀ & 5♂ = treatment 2 group	20 pigs
	Saline solution (SS)	4♀ & 4♂ = treatment 1 group	4♀ & 4♂ = control group	16 pigs
		18 pigs	18 pigs	36 pigs

Transported: road transportation for 3h; dexamethasone: 0.36 mg/kg, i.m.; and saline solution: NaCl 0.9%; i.m. 5 mL/pig.

## Data Availability

The data presented in this study are available upon request from the corresponding authors.

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
