# Peer review of "Faecal Glucocorticoid Metabolites and Hair Cortisone/Cortisol Measurements in Domestic Pigs Exposed to Road Transportation and Dexamethasone Treatment"

_animals, 2024, doi:10.3390/ani14182700_

Round 1
Reviewer 1 Report
Comments and Suggestions for Authors
The manuscript "Faecal Glucocorticoid Metabolites and Hair Cortisone/Cortisol Measurements in Domestic Pigs exposed to Road Transportation and Dexamethasone Treatment" offers helpful information about the physiological stress response in pigs during transport. The findings explore the effectiveness of fecal glucocorticoid metabolites and hair cortisol and cortisone, as well as their ratio, which is highly relevant. Reliably and robustly assessing early stress response is highly relevant for farmers and productive systems.
The authors know the area and have plenty of experience in animal studies. They wrote the manuscript very well in terms of style.
The manuscript requires some modifications to be suitable for publication:
1) Please include the latest references. For example, in lines 63-64, you mention: "few studies have explored non-invasive 63 glucocorticoid (GC) measurements in pigs." and cite two relatively old studies. Have there been no new studies in the last five years on the topic? This is recurrent throughout the manuscript. Please amend accordingly.
2) In line 125: Please detail the randomization procedure.
3) Please explain your criteria for sample size. How did you get these n numbers as suitable for the study?
4) Please provide information that the study did not pseudoreplicate experimental units, especially in figure 1. Please give the n number per group in each figure.
5) In line 170, "Sample collection and Steroid Extraction," please provide details on the feces collection.
6) Why did you assign sex as a random factor? Did you not expect sex effects in the responses?
7) I cannot see the differences you mention in the text (lines 239 - 246), particularly in Figure 2. If there were no differences, please state it explicitly in the text. Besides, a figure must stand alone. In case modifications are made, please maintain consistency with the stated in the discussion.
8) In Figure 4, SEM bars seem too high for the transported group, which may imply an effect of the Dexamethasone. Why are you discriminating between factors and did not analyze both factors together?
9) In line 284, please provide a short comment on the effect of castration on steroid levels and the incidence of sex on your measurements.
10) In lines 456 - 457, can you explain why the data is unavailable?
11) Please revise each reference in terms of format. You seem to be repeating references, for example, #20 and #52. If this is correct, reorganize the manuscript accordingly.
Best wishes,
Comments on the Quality of English Language
I have no comments regarding the quality of the English language.
Author Response
Reviewer 1
Comments and Suggestions for Authors
The manuscript "Faecal Glucocorticoid Metabolites and Hair Cortisone/Cortisol Measurements in Domestic Pigs exposed to Road Transportation and Dexamethasone Treatment" offers helpful information about the physiological stress response in pigs during transport. The findings explore the effectiveness of fecal glucocorticoid metabolites and hair cortisol and cortisone, as well as their ratio, which is highly relevant. Reliably and robustly assessing early stress response is highly relevant for farmers and productive systems.
The authors know the area and have plenty of experience in animal studies. They wrote the manuscript very well in terms of style.
RE: Thank you for your comment. We hope to contribute both to the understanding of pigs’ stress responses and to the support farmers in making informed management decisions.
The manuscript requires some modifications to be suitable for publication:
- Please include the latest references. For example, in lines 63-64, you mention: "few studies have explored non-invasive 63 glucocorticoid (GC) measurements in pigs." and cite two relatively old studies. Have there been no new studies in the last five years on the topic? This is recurrent throughout the manuscript. Please amend accordingly.
RE: Yes, there are new studies. We have revised references about non-invasive glucocorticoid measurements in pigs and we have added new studies (please see in Reference section; #13-19). Thank you for the advice.
- In line 125: Please detail the randomization procedure.
RE: A detail of the randomization procedure was added (line 135).
- Please explain your criteria for sample size. How did you get these n numbers as suitable for the study?
RE: In a repeated measures study, the experimental unit is observed at multiple time points. In our current study, we performed 5 measurements (faeces) on each pig. Following the advice of our statistics specialist, we applied the criteria n>p, where n=pigs and p=measurements.
- Please provide information that the study did not pseudoreplicate experimental units, especially in figure 1. Please give the n number per group in each figure.
RE: The term pseudoreplication refers to situations where treatments are not replicated or where replicates are not statistically independent. In our study, the treatments were applied to different pigs, so there is no risk of pseudoreplication. We added the number of pigs per treatment in figure 1.
- In line 170, "Sample collection and Steroid Extraction," please provide details on the feces collection.
RE: Details were added (lines 181-184).
- Why did you assign sex as a random factor? Did you not expect sex effects in the responses?
RE: When applying repeated measurement by mixed models, we need to define fixed and random factors. We wanted to show only differences associated with transport and dexamethasone, that´s why these sources of variation were considered as fixed factors to account for response over time. In order to account for variability among pigs, we considered sex and pig (experimental unit) as random factors.
- I cannot see the differences you mention in the text (lines 239 - 246), particularly in Figure 2. If there were no differences, please state it explicitly in the text. Besides, a figure must stand alone. In case modifications are made, please maintain consistency with the stated in the discussion.
RE: We revised the text and the figure to make them clearer and more precise.
- In Figure 4, SEM bars seem too high for the transported group, which may imply an effect of the Dexamethasone. Why are you discriminating between factors and did not analyze both factors together?
RE: On figure 4, we plotted the saline solution group vs dexamethasone group. Perhaps, it is true that within the transport group (T+SS and T+D) SEM bars are high due to the contribution of different treatments: combination of transport with solution saline or dexamethasone. Nevertheless, please note that we considered all treatments (please consider table 1) in the statistical analysis, and presented unifactorial differences in figure 2 and 3 of the revised version, since there was no bifactorial effect between transport and dexamethasone.
- In line 284, please provide a short comment on the effect of castration on steroid levels and the incidence of sex on your measurements.
RE: A brief comment was provided (lines 304-306).
- In lines 456 - 457, can you explain why the data is unavailable?
RE: Apparently, there was a confusion. Our statement indicates that the data is available. We pointed out that it is available upon request. As we are working on an internal repository at CONICET, this institution strongly recommends using its database.
- Please revise each reference in terms of format. You seem to be repeating references, for example, #20 and #52. If this is correct, reorganize the manuscript accordingly.
RE: We have checked these references (please see below) and we did not find any error. However, we did find other duplicates in the reference section, so we have carefully checked all references once more.
# 20. Sharpley et al Physiol. Res., 2009, 58, 757-761.
# 52= Keckeis, K. et al J. Comp. Physiol. B, 2012, 182, 985-996.
Best wishes,
RE: thanks so much!
Submission Date 14 August 2024
Date of this review 20 Aug 2024 12:36:11
Reviewer 2 Report
Comments and Suggestions for Authors
This study investigated pig stress responses to transportation and dexamethasone treatment: a 2×2 factorial study of faecal glucocorticoid metabolites and 60-day hair corticosteroid levels.
There may be a problem in the experimental design. I cannot understand: Why choose to collect hair samples for detection 55 days after the end of the experiment? The detection results of the hair samples also proved that the results cannot be explained. Because the intermediate time was too long, uncontrollable stress factors would affect the experimental results. The effects of experimental treatments would also fade with the extension of time.
Introduction: The authors did not mention the basis and hypothesis for hair sampling and detection after 60 days of treatments in this study.
Line 179: sampled after 60 days of treatments? Why? The 60-day hair sample can reflect the chronic or repetitive stress experienced by the animal during this period. But within 60 days after the end of the experiment, can the authors control some accidental other stress factors?
Line 190-195: “Measurement of Faecal Glucocorticoid Metabolites”, Please specify which metabolite was specifically detected?
Line 213: “Hair data were also log-10 transformed to meet normal distribution”, In Figures 2, 3, and 4, the author did not show the log-10 transformed data. Why?
Figure 2: Is there a significant difference between each group in the figure? Please mark it. Why were the cortisol levels of transported pigs even lower? How can this be explained? The hair was sampled after 60 days of treatments. Was the results still related to the treatments?
Figure 3: Why did transportation instead increase the ratio of cortisone to cortisol? Transportation has a stress effect and cortisol should be increased, so this ratio should be decreased, right? This may be because the sampling time point was on the 60th day after the transportation treatment.
Figure 4: After injecting dexamethasone, cortisol levels usually decrease. Dexamethasone is a long-acting glucocorticoid with a strong inhibitory effect on adrenal cortical function. It inhibits the hypothalamic-pituitary-adrenal axis (HPA axis) through a negative feedback mechanism, thereby reducing the secretion of endogenous cortisol. Then the ratio of cortisone to cortisol should increase, right?
The results of your Figures 2, 3, and 4 are all contrary to common sense. These results may be because the hair sampling time point was on the 60th day after the transportation treatment. The selection of this sampling time point for pig hair two months after treatment is very strange. The authors did not explain it in the Introduction, nor did they explain it well in the Discussion.
Line 276-280: Here, there is no discussion of the reasons and no mention of the sampling time point. As for the cortisone/cortisol ratio, the author said it confirmed their predictions, but I did not see the author's relevant hypothesis or predictions in the Introduction or elsewhere.
Author Response
Reviewer 2
Comments and Suggestions for Authors
This study investigated pig stress responses to transportation and dexamethasone treatment: a 2×2 factorial study of faecal glucocorticoid metabolites and 60-day hair corticosteroid levels.
RE: Thank you for your comment. We hope to contribute both to the understanding of pigs’ stress responses and to the support farmers in making informed management decisions.
There may be a problem in the experimental design. I cannot understand: Why choose to collect hair samples for detection 55 days after the end of the experiment? The detection results of the hair samples also proved that the results cannot be explained. Because the intermediate time was too long, uncontrollable stress factors would affect the experimental results. The effects of experimental treatments would also fade with the extension of time.
RE: To our best knowledge, there is still no period defined between hair shaving and quantification of glucocorticoids in this type of samples. After reviewing several reports, we chose a sampling period previously tested by Heimbürge et al 2020(GCE., https://doi.org/10.1016/j.ygcen.2019.113359), who measured pig hair cortisol at 28 and 56 days post-treatments.
Regarding sampling timing, current reports use hair cortisol as a retrospective measure of a previous period, but, as far as we know, there is no gold standard for samplings, applicable to both humans and other mammals. Generally, hair grows approximately 1 cm per month in mammalian species, but Heimbürge et al 2020 found hair growth in pigs’ neck of up to 5 mm per month. So, after shaving, the part of the hair present within (and close above) the skin needs some time to grow above the surface to be cut. Based on the 5 mm/month, we chose the 60 days to get enough hair after reshaving for measurement. We improved introduction to support better the experimental design (lines lines 90-94).
Furthermore, to address confounding factors in sampling time and the concern about the fading of effects of treatments over time, we followed recommendations from previous studies in pigs to minimize this situation, since the mechanism of cortisol deposition in hair is still unclear. Heimbürge et al.(2020) recommended: “comparing results only from animals of the same age, group, hair samples taken from the same body region and the same hair color. In addition, the age of hair segments seems to be a confounding factor that cannot be controlled or standardized easily as other influencing factors. It is advisable to use regrown hair from previously shaved areas for the reflection of a particular time period. This approach will also help to minimize the confounding effect of cortisol accumulation in older hair segments, which may be caused by contamination”. In our study, all pigs were exposed to the same treatments, environmental conditions, and hair collection methodology.
Introduction: The authors did not mention the basis and hypothesis for hair sampling and detection after 60 days of treatments in this study.
RE: We previously mentioned some basic information about hair detection in the former version (lines 108-109). Considering your comments, we added a brief comment about hair sampling (lines 90-94).
Line 179: sampled after 60 days of treatments? Why? The 60-day hair sample can reflect the chronic or repetitive stress experienced by the animal during this period. But within 60 days after the end of the experiment, can the authors control some accidental other stress factors?
RE: We added additional information in the revised version, as we pointed out in our previous reply to your comment above. Besides, although we did not control other stress factors within the 60-day period, all the studied pigs were reared in the same unit and under identical management conditions. Hence, any potential “noise” in the experimental design would have affected all pigs equally. Technically, the studied pigs were housed under controlled environmental conditions, with no human intervention in the study, such as group mixing, change of pens, transportation, or veterinary procedure. Pig keepers did not report any unexpected situation that presumably could have affected stress levels of these animals. Therefore, we consider that the influence of unexpected stress factors was negligible in the present study.
Line 190-195: “Measurement of Faecal Glucocorticoid Metabolites”, Please specify which metabolite was specifically detected?
RE: In all animal species, several FGM are excreted in the feces (their exact identity is mostly unknown). We state the exact name of the standard of the EIA. This assay is a group-specific EIA, which measures metabolites with a 5a-3ß,11ß-diol configuration. We added this info to the text (lines 203-205)
Line 213: “Hair data were also log-10 transformed to meet normal distribution”, In Figures 2, 3, and 4, the author did not show the log-10 transformed data. Why?
RE: Our statistic specialist has pointed out that it is acceptable to present both transformed and untransformed data. However, we prefer to show untransformed data to emphasize the biological relevance of the measurements (e.g. the mass of hormone quantified by EIA).
Figure 2: Is there a significant difference between each group in the figure? Please mark it. Why were the cortisol levels of transported pigs even lower? How can this be explained? The hair was sampled after 60 days of treatments. Was the results still related to the treatments?
RE: Considering your comments and those of Reviewer 1, we revised the text (lines 253-257) and modified figure 2, briefly. We improved the text and added information about significant differences between groups in figure 2. Right now, we don´t have an explanation about higher values of hair cortisol in the not transported group than in the transported group. The unexpectedly high values in hair cortisol detected in the control group compared to the values detected in transported pigs, as measured by the cortisol EIA, may be attributed to different confounding factors. We speculated that the hair sampling period and/or the climatic conditions, rather than a lack of effect of biological and/or pharmacological challenges, would have a confounding effect on our findings. We write about this finding in the discussion (lines 400-431).
Figure 3: Why did transportation instead increase the ratio of cortisone to cortisol? Transportation has a stress effect and cortisol should be increased, so this ratio should be decreased, right? This may be because the sampling time point was on the 60th day after the transportation treatment.
RE: We believe that our discussion (former lines 362-374; now 384-399) should be useful to explain these concerns. Briefly, we agree that transportation has a stressing effect on pigs, which would increase cortisol levels in the bloodstream. However, apparently, most cortisol is not incorporated into hair after stressful situations, as was suggested for guinea pig after a cortisol radiometabolism study. Therefore, perhaps also in the present study we quantified hair cortisol from local production (follicle), and cortisol from systemic blood was converted to cortisone in the hair, generating an increase in the value of the numerator (cortisone) and a lower or constant value of the denominator (cortisol), in the cortisone/cortisol ratio.
Figure 4: After injecting dexamethasone, cortisol levels usually decrease. Dexamethasone is a long-acting glucocorticoid with a strong inhibitory effect on adrenal cortical function. It inhibits the hypothalamic-pituitary-adrenal axis (HPA axis) through a negative feedback mechanism, thereby reducing the secretion of endogenous cortisol. Then the ratio of cortisone to cortisol should increase, right?
RE: Similarly to our reply above, we agree with you about the effect of dexamethasone on HPA axis. However, considering that systemic cortisol may not be incorporated into hair because it is converted to cortisone, we expected to find lower values of cortisone/cortisol ratio in dexamethasone-treated pigs than in pigs treated with saline solution. In other words, dexamethasone inhibits adrenocortical activity, reducing systematic cortisol levels; therefore, we would expect to find less cortisone in hair after this conversion.
The results of your Figures 2, 3, and 4 are all contrary to common sense. These results may be because the hair sampling time point was on the 60th day after the transportation treatment. The selection of this sampling time point for pig hair two months after treatment is very strange. The authors did not explain it in the Introduction, nor did they explain it well in the Discussion.
RE: The fact that the mechanism of blood cortisol deposition in hair is still not fully known and may lead to inconsistencies in sampling periods reported in the literature, as well as in our results, which could be reflected in figure 2. Perhaps sampling time is a factor affecting imprecision, but different studies can contribute to the present puzzle. However, we believe that Figures 3 and 4 are compatible with the previous radiometabolism report, which indicated that cortisol is converted to cortisone. The combination of these measurements supports the expected results. We agree with you that an explanation was missing in the introduction ; therefore, we have now added relevant details (lines 86-90). We also revised the English style in our discussion to avoid misleading interpretation (former lines 362-374, now lines 384-399).
Line 276-280: Here, there is no discussion of the reasons and no mention of the sampling time point. As for the cortisone/cortisol ratio, the author said it confirmed their predictions, but I did not see the author's relevant hypothesis or predictions in the Introduction or elsewhere.
RE: We have discussed the topic of the sampling point (lines 400-415; former version, lines 379-394). You are right about the lack of mention of cortisone/cortisol ratio. We have briefly addressed our predictions regarding this ratio in introduction (lines 117-118).
.
Submission Date
14 August 2024
Date of this review
23 Aug 2024 04:22:49
Round 2
Reviewer 1 Report
Comments and Suggestions for Authors
The manuscript improved and it is suitable for publishing. Only two last comments:
Please improve the quality of Figure 2.
Please include a justification for the decision for your sample size.
Author Response
Reviewer 1/Round 2
Comments and Suggestions for Authors
The manuscript improved and it is suitable for publishing. Only two last comments:
RE: Thank you for your comment. We are glad to know the manuscript improved. We addressed all concerns, please see below.
Please improve the quality of Figure 2.
RE: Please note that we have changed Figure 2, 3 and 4 by suggestion of reviewer 2, and also improved quality. Since the statistical analysis was made based on log 10 transformed data reviewer 2 asked to show the log 10 transformed data on the figures 2, 3 , 4
Please include a justification for the decision for your sample size.
RE: We included a brief statement for the decision, see lines 137-139.
Submission Date
14 August 2024
Date of this review
06 Sep 2024 13:44:49
Reviewer 2 Report
Comments and Suggestions for Authors
After the author's revision, the article has been improved. But there are still some concerns that need to be addressed by the authors.
Line 190-194: Please describe the length and segment of the sampled hairs. The hairs collected are only meaningful if they were grown during the experimental period.
Line 225-226: If this data is not normally distributed, I would suggest that you can use non-parametric tests for statistics that do not require the data to be normally distributed. If a logarithmic transformation is to be performed, then the significance results obtained are for the log-transformed data. These are not consistent with the results you show in Figures 2, 3, and 4.
The results of your Figures 2, 3, and 4: The composition of the hair is fixed once it grows out, so the cortisol content of different sections of hair is representative of the amount absorbed during growth. As you mentioned the study of Heimbürge et al 2020 (GCE., https://doi.org/10.1016/j.ygcen.2019.113359), Different segments of the same hair have different levels of cortisol. Therefore, it becomes important to know which segment of the hair was measured as a result of Figures 2, 3, and 4.
How long were the effects of these experimental treatments? If the growth rate of pig hair was 5 mm/month, how many cm did the hairs grow during this period? Only then do the results of the study make sense when they correspond. Otherwise the results of Figures 2, 3 and 4 are not very relevant to the subject.
Author Response
Reviewer 2/ round 2
Comments and Suggestions for Authors
After the author's revision, the article has been improved. But there are still some concerns that need to be addressed by the authors.
RE: Thank you for your comments. We are glad to know the manuscript improved. We addressed all concerns, please see below.
Line 190-194: Please describe the length and segment of the sampled hairs. The hairs collected are only meaningful if they were grown during the experimental period.
RE: We agree with you; collected hair should include the hair which was grown (or exposed: diffusion of GCs into the hair is also possible) during the experimental period. As we needed enough material for hair cortisol analysis, we waited the 60 days, during which the hair present in the experimental period (within the skin) grew outside the skin and could be removed (see figure below). We added a brief statement that hair length was ~ 10 mm (see lines 196-198).
For a detailed explanation of pigs hair collection in our study we edited Fig 2 published by Heimbürge et al 2020 (GCE., https://doi.org/10.1016/j.ygcen.2019.113359, please see below (OR YOU CAN SEE THE ATTACHED DOCUMENT). You can see different hair segments in the “original hair” picture. First, we shaved the hair before the start of experimental period (see shaved picture; red: hair present at that time). The hair remaining within the skin (and the freshly growing one) during the experimental phase was exposed to circulating GCs. Secondly, we re-shaved (see Re-shaved picture) hair on the same area to collect hair (red part now present above the skin) for glucocorticoid measurements. Of course, it is hard to collect only hair, which was present during the experimental phase, but this is a limitation of hair as sample material for measuring acute stress, and we tried our best to achieve it.
Line 225-226: If this data is not normally distributed, I would suggest that you can use non-parametric tests for statistics that do not require the data to be normally distributed. If a logarithmic transformation is to be performed, then the significance results obtained are for the log-transformed data. These are not consistent with the results you show in Figures 2, 3, and 4.
RE: you are right, so, we replaced figures 2, 3 and 4 to show log 10 transformed data.
The results of your Figures 2, 3, and 4: The composition of the hair is fixed once it grows out, so the cortisol content of different sections of hair is representative of the amount absorbed during growth. As you mentioned the study of Heimbürge et al 2020 (GCE., https://doi.org/10.1016/j.ygcen.2019.113359), Different segments of the same hair have different levels of cortisol. Therefore, it becomes important to know which segment of the hair was measured as a result of Figures 2, 3, and 4.
RE: considering our reply for lines 196-198 (above), and hope that this comment was addressed.
How long were the effects of these experimental treatments? If the growth rate of pig hair was 5 mm/month, how many cm did the hairs grow during this period? Only then do the results of the study make sense when they correspond. Otherwise the results of Figures 2, 3 and 4 are not very relevant to the subject.
RE: The effect of transport and dexamethasone on adrenocortical activity was an acute one. As shown in Fig 1, faecal cortisol metabolites were elevated shortly after transportation and lasted 36 hs. Plasma GCs are thought to diffuse into the hair, thus the hair present during the experiment (within the skin) was exposed to higher cortisol (and due to conversion - see the discussion) also cortisone levels. Enough time is needed to allow exposed hair to grow above the surface and to collect enough hair for GC analysis. Due to a “pooling” effect, acute stressors need to be strong to be visible, which is a limitation of hair samples in acute stress situations and probably one of the reasons why we could not see an effect in hair cortisone or cortisol concentrations. However, we think that it is also important to report negative findings, and interestingly, we found that the cortisone/cortisol ratio may be better suited to track such changes.
Submission Date
14 August 2024
Date of this review
03 Sep 2024 04:42:06
